# Thermoelectric and Structural Properties of Transparent Sb-Doped ZnO Thin Films Sputtered in a Confocal Geometry

Helder Filipe Faria [1], Joana Margarida Ribeiro [1], Torben Boll [2] and Carlos José Tavares [1,*]

1 Physics Centre of Minho and Porto Universities (CF-UM-PT), University of Minho, 4835-386 Guimarães, Portugal

2 Institute for Applied Materials (IAM-WK), Institute for Nanotechnology (INT), Karlsruhe Nano Micro Facility (KNMFi), Karlsruhe Institute of Technology (KIT), D-76344 Karlsruhe, Germany

* Correspondence: ctavares@fisica.uminho.pt; Tel.: +351-253510474

**Abstract:** This study focuses on understanding the influence of low Sb doping on ZnO's electrical, optical, and thermoelectrical properties, while also studying its structural and morphological parameters. For this, several ZnO films with varying Sb target current densities, in the range of 0–0.27 mA/cm$^2$, were produced by DC magnetron sputtering in a confocal geometry. As a result, thin ZnO:Sb films with an average transparency in the visible region greater than 80% are obtained, revealing for optimized conditions an absolute Seebeck coefficient of 100 μV/K and a respective power factor of 1.1 mW·m$^{-1}$·K$^{-2}$ at 300 K, effectively modifying the electrical, optical, and thermoelectrical properties of the material and ensuring its suitability for heat harvesting applications. From atom probe tomography experiments, a larger Zn content is registered at triple junctions of the grain boundary, which matches the approximately 25 nm crystallite grain size derived from the X-ray diffraction analysis.

**Keywords:** ZnO; antimony; Seebeck coefficient; thermoelectric; thin films; magnetron sputtering; atom probe tomography





## 1. Introduction

Energy is and will continue to be a crucial part of the everyday life of our society. According to the Annual Energy Outlook 2021 made by the EIA [1], by 2050, global energy use is expected to increase by nearly 50% compared to 2020. Similarly, the World Energy Outlook 2021 [2] found evidence supporting the energy demand growth and its effect on carbon emissions. As renewable energies still lack the potential to sustain the demand, other types of solutions have gained importance, namely heat. Heat is present in almost every system, and it has been shown to be responsible for a large portion of all wasted energy [3].

There are several types of heat-related systems, some of which utilize reutilized heat that is dissipated during industrial processes, and some that rely on direct electrical conversion, such as thermoelectric systems. Thermoelectric materials transform heat into electricity (or vice versa) and have drawn much attention as heat-harvesting sources. This technology is based on the Seebeck effect [4], and for a material to be considered a good thermoelectric, it has to have large Seebeck coefficient for maximum electricity conversion, large electrical conductivity, and small thermal conductivity. These are all factors that influence the performance of the material, and are equated by the figure of merit (ZT), which is a dimensionless quantity that represents the thermoelectric quality of the material, and is given by the following [4]:

$$ZT = \frac{S^2\sigma}{k}T \tag{1}$$

where $T$ is the operating temperature in kelvin (K), $S$ is the Seebeck coefficient (V/K), $\sigma$ is the electrical conductivity ($\Omega^{-1}\cdot cm^{-1}$, or $S\cdot cm^{-1}$), and $k$ is the thermal conductivity ($W\cdot m^{-1}\cdot K^{-1}$).

Alternatively, the thermoelectric power factor (PF, in $W\cdot m^{-1}\cdot K^{-2}$) [4] can also be used to quantify the material in cases where the thermal conductivity has not been measured or calculated, as is the present case.

$$PF = S^2\, \sigma \tag{2}$$

Several types of materials, including organic and inorganic, have been used in thin film technology as thermoelectric materials, as they have significant benefits when compared to their less efficient and more expensive bulk forms [5–7]. The low dimensional material design was proved to improve thermoelectric efficiency through more effective phonon scattering, contributing to a reduced lattice thermal conductivity, and, at the same time, increasing the PF [4], popularizing thin films and nanostructures. Studies on bismuth telluride, a popular thermoelectric material, achieved a PF of 2.5 $mW\cdot m^{-1}\cdot K^{-2}$ (at 473 K) and 1.2 $mW\cdot m^{-1}\cdot K^{-2}$ (at room temperature, RT) in lead-doped and flexible undoped $Bi_2Te_3$ thin films, respectively [6,7]. However, although being very efficient as thermoelectrics, these materials are opaque and cannot be applied in optically transparent devices. Metal oxides have potential as transparent thermoelectrics. Several experimental studies on $TiO_2$- and ZnO-based thermoelectric films have reported interesting thermoelectric performance, albeit with smaller power factors compared with the aforementioned bismuth telluride. For example, $TiO_2$:Nb 150 nm thick films with a PF of 500 $\mu W\cdot m^{-1}\cdot K^{-2}$ at 300 K (ZT = 0.18) [8]; ZnO:Ga,Bi and ZnO:Al,Bi 300 nm thick films with a PF in the range of 70–80 $\mu W\cdot m^{-1}\cdot K^{-2}$ at 300 K [9]; ZnO:Al 150 nm thick films with a PF of ~200 $\mu W\cdot m^{-1}\cdot K^{-2}$ at 510 K [10]; ZnO:Ti 100 nm thick films with a PF of ~1 $\mu W\cdot m^{-1}\cdot K^{-2}$ at 473 K [11]; ZnO:S films with a PF of ~5 $\mu W\cdot m^{-1}\cdot K^{-2}$ at 310 K [12]; ZnO:Sb 600 nm thick films with a PF of ~400 $\mu W\cdot m^{-1}\cdot K^{-2}$ at 700 K [13]; p-type ZnO:Sb 500 nm thick films with a PF of ~50 $\mu W\cdot m^{-1}\cdot K^{-2}$ at 773 K.

In addition to electrical conductivity, some materials also possess high transparency in the visible range. These materials are denominated as transparent conducting oxides (TCOs) and have been investigated for a wide range of applications in optoelectronic devices including photovoltaics, displays, transparent electrodes in touch panels, optoelectronic interfaces, and window glass technologies [14,15].

The design of a TCO material with thermoelectric properties would enable the production of electrical energy through heat harvesting from the environment in all kinds of transparent devices. Even though a wide range of thermoelectric materials have been studied and explored, a large ZT value remains connected to bulk materials [16]. The problem is that, because of their small band gap, these types of materials tend to be optically opaque, not constituting a valid solution to the intended applications [17]. Therefore, a material that provides all the necessary characteristics of a TCO with good thermoelectric properties is still necessary to allow for practical applications.

Zinc oxide (ZnO) is a well-known inorganic compound quite common in modern electronics, mostly due to its abundance and versatility. It has a wide direct band gap of 3.44 eV at low temperatures and 3.37 eV at room temperature [18]. It crystallizes in the hexagonal wurtzite structure, displaying an intrinsic n-type conductivity, although in recent years it has become a target of interest due to its semiconductor properties as researchers try to control the unintentional n-type conductivity and achieve p-type conductivity [19]. Furthermore, added to the possible conductivity characteristics, the optical properties of ZnO make it a viable solution for TCO technology, but its relatively large electrical resistivity is a constraint that is usually countered by doping. Usually, n-type doping of ZnO is achieved with Group-III elements, such as aluminum and gallium, that can act as substitutional elements for Zn and group-VII elements, such as chlorine and iodine, that can serve as substitutional elements for oxygen [20–23]. As for p-type doping, is not as straightforward as n-type doping because it is very difficult to obtain p-type doping in wide band gap semiconductors.

Other Group-V elements have been studied, and a promising yet also commonly researched element with good potential is antimony. Several different studies focused on and effectively achieved p-type conductivity, with small electrical resistivity, in Sb-doped ZnO thin films, through a variety of deposition methods, including molecular-beam epitaxy, pulsed laser deposition, RF magnetron sputtering, spray pyrolysis, and oxidative evaporation [24–28]. Still, the most successful study, which used an approach based on sol–gel and hydrothermal methods, achieved an electrical resistivity of 2.1 mΩ·cm [29]. These studies effectively show the electrical potential of ZnO:Sb thin films, but studies on their thermoelectric properties are still scarce.

The novelty of this study is in the fabrication of thermoelectric ZnO:Sb thin films by confocal sputtering, with TCO characteristics and the complete evaluation of their optical, electrical, and thermoelectrical properties.

## 2. Materials and Methods

Undoped and Sb-doped ZnO thin films were prepared by a custom-made DC magnetron sputtering system conceived at the Centre of Physics of the University of Minho (see Figure A1 for the setup's schematic). A confocal geometry is associated with two circular magnetrons, each with a diameter of 10 cm. One had a target of ZnO with 99.99% purity fabricated by FHR Anlagenbau GmbH (Saxony, Germany), and the other had an Sb target also with 99.99% purity supplied by Photon Export. A series of depositions were executed but, before each one, the chamber was evacuated with a primary rotary pump and a turbomolecular pump to achieve a base pressure of around $10^{-4}$ Pa. Each set of films, for a specific level of Sb doping, were deposited on $76 \times 26 \times 1$ mm$^3$ glass substrates from Normax Lda (Leiria, Portugal), along with four $10 \times 10 \times 0.5$ mm$^3$ Si substrates, cut from P/B doped Si-wafer <100>, from Siegert Wafers GmbH (Aachen, Germany), following the schematic in Figure A2. Before assembling the substrates inside the chamber, they were cleaned with isopropyl alcohol (2-propanol) and acetone in an ultrasonic bath, both for 15 min, removing contaminants on the surface of the substrates that may arise during storage and handling, in order maximize the adhesion at the film/substrate interface. The substrate holders were mounted onto a support controlled by a motor that operated in continuous rotation during deposition to ensure uniformity. The distance between the targets and the substrate was kept at 8 cm, and the substrate holder was heated before and during depositions. Research-grade Ar with 99.999% purity was used as a working gas to enable plasma formation. Ahead of every deposition, 3 min of ion etching at 500 V was performed in an Ar atmosphere at a pressure of 1.6 Pa. This process is used to remove oxides and impurities that possibly accumulated and were retained on the surface of the substrates, which could create atomic defects in the substrate, enabling better nucleation of the film during deposition.

The series of depositions started with a dummy deposition of 2 h for cleaning (burn) of the ZnO target, which was new. This was followed by a control deposition where the Sb target had no current and was covered with an Al sheet to prevent unwanted interactions with the plasma. Then, a series of depositions, of 35 min each, were performed where the current applied in the Sb target was gradually increased with steps of 2 mA, starting from 0 mA in the first deposition, 1 mA for the second, and ending with 21 mA (in the range of 0–0.27 mA/cm$^2$). This was the only parameter that suffered variations, as all others were kept the same between depositions, intending to effectively study how it would change doping levels and how that change would affect the thin film properties. Already tested deposition process parameters used in other studies in the same setup were followed. Target current density, bias voltage, voltage applied in the ZnO target, and Ar flow were fixed at 10 mA·cm$^{-2}$, −60 V, 400 V, and 40 sccm, respectively, maintaining a working pressure between 0.18 and 0.21 Pa. Optically transparent and conducting films were effectively fabricated, and their properties were accordingly studied.

An in-depth study of the electrical properties of the ZnO:Sb thin films was carried out based on the Hall effect, utilizing an Ecopia AMP55T HMS-5000 Hall effect measuring

system, with a DC four-point probe apparatus in the Van der Pauw configuration. The Hall voltage (VH) was determined, and the respective carrier concentration (*n*) and mobility (μ) were calculated with the following Equations (3) and (4) [30]:

$$n = \frac{|B| \cdot I}{e \cdot |VH| \cdot t} \tag{3}$$

$$\mu = \frac{1}{e \cdot \rho \cdot n} \tag{4}$$

where B is the magnetic field, which was kept at a constant 0.560 T, t is the film thickness, I is the injected current, and ρ is the electrical resistivity. Ten measurements were performed for each sample deposited on Si substrates, and an average was computed for each of the significant properties referred above, in an attempt to obtain a more accurate result.

Values for optical transmittance and reflectance were acquired through measurements with a Shimadzu UV-2501PC UV–Vis spectrophotometer. The samples deposited on glass substrates were analyzed in the 300 to 900 nm wavelength range and the average for each quantity was estimated between 400 and 700 nm for posterior calculations and analysis. For both studies, a baseline was established using two samples of glass without any film deposited on them, and then one was kept as a reference when studying each sample. Finally, with the transmittance spectra and custom-made software called *dCalc*, the thickness and optical properties of each sample were calculated. The software utilizes interference patterns to determine the refractive index of the film and, consequently, its thickness, and follows the Swanepoel method to determine the optical properties, namely the absorption index and band gap [31].

A study on the morphology and cross-section of the thin films was performed with scanning electron microscopy (SEM), performed with an FEI NOVA NanoSEM 200. To perform cross-section imaging, the samples were cut with a diamond tip and fixed in the sample table with double-sided carbon tape.

Atomic force microscopy (AFM) was also used to study the surface properties of the films, namely roughness, on a CSI Instruments Nano-Observer atomic force microscope. For each sample, two different zones of the film were analyzed, typically one near the center and one near the edge. A mean of both values was obtained afterwards for a more accurate evaluation.

The surface chemistry of the produced ZnO:Sb films was studied with an XPS spectrometer (Kratos Axis-Supra instrument, at 3 Bs Group, University of Minho) equipped with a monochromatic Al-Kα X-ray radiation source (1486.6 eV) operated at an X-ray power of 225 W. The photoelectron spectra were collected at the take-off angle of 90° with the sample surface by means of a hemispherical electron energy analyzer operated in the constant analyzer energy lens mode (CAE). The pass energies of 160 eV and 40 eV were used for the survey and high-resolution spectra, respectively. The binding energy was referenced by setting the binding energy of the C1s hydrocarbon peak (the most intensive component of the C1s spectrum) at 284.8 eV. An electron flood gun was used to compensate for surface charging effects. Furthermore, to improve the composition study, energy-dispersive X-ray spectroscopy (EDX) was performed in an EDAX- Pegasus X4M.

X-ray diffraction (XRD) analyses were performed to study the crystallographic structure of the thin films using a Bruker AXS D8 Discover diffractometer operated in θ–2θ geometry with CuKα radiation. XRD patterns were obtained with a step size of 0.02° and an integration time of 1.5 s. Afterwards, with the help of the software *Fityk*, the XRD diffraction patterns were analyzed.

Atom probe tomography (APT) was employed to study the morphology of the grain boundaries of the thin films. The experiments were performed at the KNMFi Laboratory for Microscopy and Spectroscopy at the Karlsruhe Institute of Technology, Germany. The thin films were deposited onto a specialized silicon coupon with prefabricated microtips from CAMECA, and subsequently prepared for APT analysis using a Zeiss Auriga 60 Dual

Beam Focused Ion Beam (FIB) in order to obtain a small tip with an apex diameter of 60 nm, retaining a roughly 300 nm thick layer on top of the Si substrate. The tips were measured in a local electrode atom probe (LEAP 4000X HR) in UV-laser mode (30 pJ), with a pulse frequency of 125 kHz, at a temperature of 50 K and a detection rate of 0.4%. The data was reconstructed using the Ivas 3.6.14 software from CAMECA.

Finally, the Seebeck coefficient was evaluated using custom-made equipment that consisted of three main parts: the chamber, the pump, and a custom-made controlling software and hardware system. The pump ensured a low vacuum of approximately 1 Pa inside the chamber during the experiments. Inside the chamber were placed the holding system and the temperature control setup. Two Peltier devices (Quick-Ohm Kupper & Co. GmbH, Wuppertal, Germany) operate as a heater and cooler for the edges of the sample, applying their effect in a $25 \times 75$ mm$^2$ area. These devices ensure the difference in temperature necessary for calculating the Seebeck coefficient, and the potential difference is measured using a two-probe contact geometry, proven efficient even for higher temperatures [32]. For each sample, the thermal gradients applied on the films deposited on glass substrates varied between 30 and 55 °C with a step of 5 °C, resulting in six measurements for increasing $\Delta$T. For each measurement, 10 min of thermal stabilization was allowed to obtain a good and coherent value. These values are controlled and registered by the third part of the system, the hardware and software, where a custom-made interface is responsible for controlling the applied temperature in each Peltier device, controlling its evolution throughout the analysis, and registering each respective potential difference. After the experiment, and with all the values registered, the obtained $\Delta$V values were plotted as a function of $\Delta$T and linearly fitted to determine the slope, and subsequently the Seebeck coefficient, similar to other applications of this same system [9].

## 3. Results and Discussion

In total, 13 samples were deposited for this study. The first one, an undoped ZnO thin film (ZnOX) served as a control sample, as it was deposited with the Sb target covered, so it would not affect the normal deposition of ZnO. Then, the rest of the samples were deposited with optimized conditions and an increasing target current was applied to the Sb target, in order to produce ZnO:Sb films with varying doping levels. These depositions allowed the study of the influence of Sb doping on the structural, optical, and electrical properties of the ZnO thin films.

For the sake of simplicity, the title of each film corresponds to the current applied in the Sb target during deposition (ZnO:SbXX = the film deposited with XX current applied to the Sb target, in mA).

### 3.1. Optical Properties

As previously mentioned, a study on the optical transmittance and reflectance of each sample was performed between 300 and 900 nm (see Figure A3 for the full spectra). Then, an average value between 400 and 700 nm for each measurement is calculated and plotted in Figure 1. The results show that the increase in Sb doping, caused by the larger applied current to the Sb target, generates a slight decrease in the average transmittance, but the smallest value remains larger than 80%, so all samples have good optical transparency. As for reflectance curves, there is no relevant change, as it seems that it is independent of the doping level. Hence, even with the presence of Sb, the films preserve the natural good transparency of ZnO in the Vis and UV, which represents one of the key factors of a good TCO material.

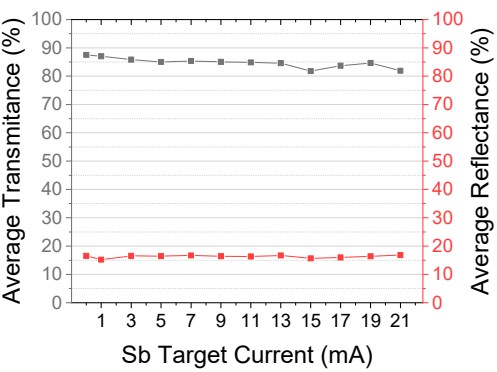

**Figure 1.** Average transmittance and reflectance as a function of the current applied to the Sb target.

*3.2. Compositional and Structural Analysis*

The control (undoped) sample ZnOX and the doped samples ZnO:Sb00, ZnO:Sb01, ZnO:Sb11, and ZnO:Sb21 were subjected to SEM analysis. As result, several micrographs of the cross-sections and of the surfaces of each sample are obtained (see Figure 2) and, with the data, an average thickness of the studied films is calculated. Afterwards, with the transmittance spectra, the thickness of each film is calculated with the software *dCalc*. Both cases showed that a greater current density in the Sb target resulted in films with a larger thickness, as it is expected. Furthermore, both studies coincided in the values they presented, with a small difference in only a few samples that never reached 100 nm. The thinnest is the control sample that showed 578 nm of thickness, and the thickest is the ZnO:Sb21 sample with 790 nm of thickness.

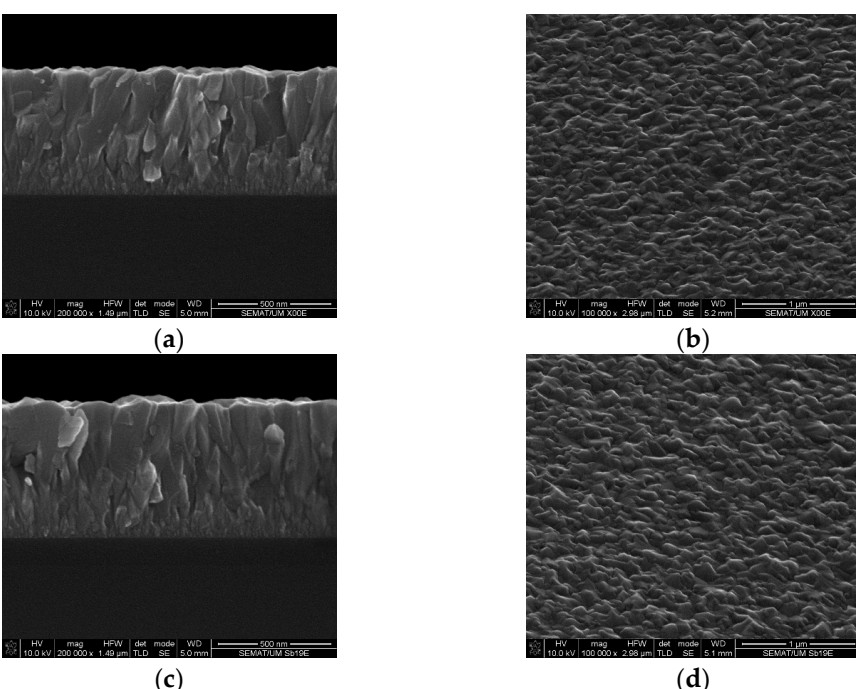

**Figure 2.** *Cont.*

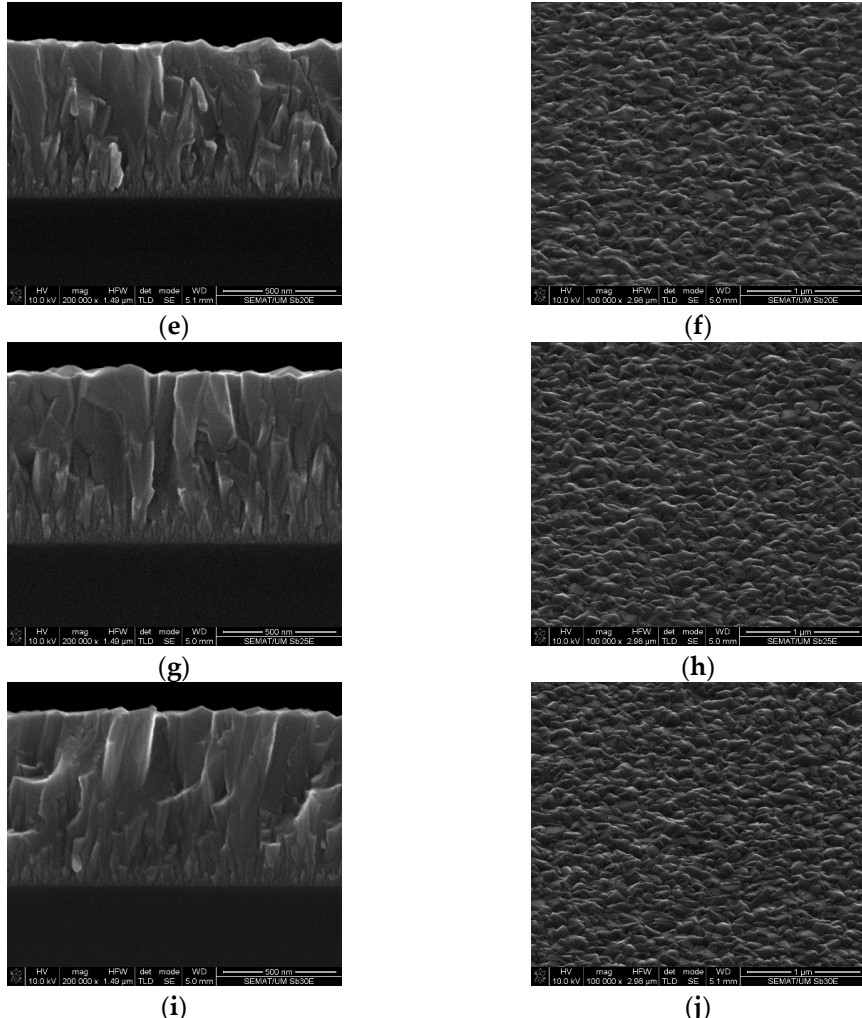

**Figure 2.** SEM micrographs of the cross-sections and surfaces of the (**a**,**b**) ZnOX, (**c**,**d**) ZnOSb:00, (**e**,**f**) ZnO:Sb01, (**g**,**h**) ZnO:Sb11, and (**i**,**j**) ZnO:Sb21 samples. The scale bar is 500 nm and 1 μm for the micrographs on the left and right, respectively.

The *dCalc* software also produced the Tauc plot of each sample, estimating the direct energy band gap. The effect of the Sb doping is reflected in this analysis, as a decrease from 3.37 eV for the undoped ZnO to 3.25 eV for the Sb-doped ZnO is registered, without any significant change between doped samples.

Then, the same samples were subjected to EDX analysis. We expected to see an increase in the atomic concentration of Sb with the increase in the current applied to the Sb target, meaning that the doping level is larger, but this did not happen. The concentration of Sb in all doped films is approximately 0.5 at.%. This must be related to the fact the limit of solubility of Sb in the wurtzite structure is around ~0.5 at.%, and a further increase in the Sb target current density in confocal geometry does not overcome this threshold. The absence of Sb in the control sample verifies that covering the target prevented impurities.

Next, an XPS analysis is performed, and the fitted spectra for the Zn 2p (doublet), O 1s (singlet), and Sb 3d (doublet) core line levels for an undoped sample (ZnOX) and a ZnO:Sb sample (ZnO:Sb21) are studied (Figures 3 and 4). The composition is approximately stoichiometric for the undoped sample, with a slight deficiency in Zn (48.7 at.%). It should be noted that XPS is only sensible to the topmost atomic layers (~7 nm below the surface); hence, this slight zinc deficiency may be attributed to some evaporation of this material, which competes with sputtering due to its low melting temperature. For the case of the Sb-doped sample, the sub-stoichiometry is reversed, with the film being deficient in oxygen (48.8 at.%), possibly due to the promotion of oxygen vacancies upon Sb

doping (1.0 at.%) in the wurtzite structure; further photoluminescence experiments needed to be performed to confirm these vacancies. The latter value of Sb doping is larger than that is measured by EDX (0.34 at.%, in Table A1), since EDX averages the film bulk, while XPS only considers the topmost atomic layers; hence the registered increase in Sb at the surface is possibly caused by diffusion processes. From Figures 3a and 4a, it can be seen that the shape and position of the Zn $2p_{3/2}$ peak (1021.3 eV) is practically unaltered with doping, with the doublet separation being the same ($\Delta E = 23.1$ eV), and the peaks full width at half maximum (FWHM, $\beta$) are within the range of 1.5–1.7 eV. However, the impact of Sb doping is readily viewed for the O 1s core line (Figures 3b and 4b). Since the O 1s core line overlaps with the Sb $3d_{5/2}$ main doublet peak, the fitting of these two core lines presents a challenge. The used strategy is to first fit the Sb 3d doublet (Figure 4c) and, from the position of the lower intensity doublet peak (Sb $3d_{3/2}$), the relative areas between the $3d_{5/2}$:$3d_{3/2}$ (3:2) doublet peaks, the determined spin-orbit splitting (9.6 eV) of these peaks, the deduced position of Sb $3d_{5/2}$ (530.2 eV), and its FWHM (1.2 eV), it is then possible to fit the O 1s core line (Figure 4b) by forcing the insertion of the $3d_{5/2}$ in its envelope structure (component CSb in Table 1 and Figure 4c). From the NIST X-ray Photoelectron Spectroscopy Database [33] it can be derived that the average binding energies for Sb metal, $Sb^{3+}$ in $Sb_2O_3$, and $Sb^{5+}$ in $Sb_2O_5$, are, respectively 528.04 eV, 530.13 eV, and 531.45 eV. Since from Table 1 and Figure 4 the binding energy ascribed to the Sb 3d5/2 is 530.2 eV, it can be concluded that the valence state of Sb in the ZnO wurtzite structure is 3+. For both samples, the O 1s core line is fitted with two contributions as follows. For the smaller binding energy (529.9–530.0 eV) component C1 is ascribed to Zn-O bonds and Sb-O bonds (doped sample), and its position only varies marginally (0.1 eV), with unaltered FWHM. Conversely, component C2 (531.6–532.0 eV) is attributed to adsorbed oxygen and OH groups, mostly from surface contamination, and C2 is not taken into account to determine film composition. The area of C1 is slightly larger (66%) for the undoped sample, in comparison to the Sb-doped sample (63%).

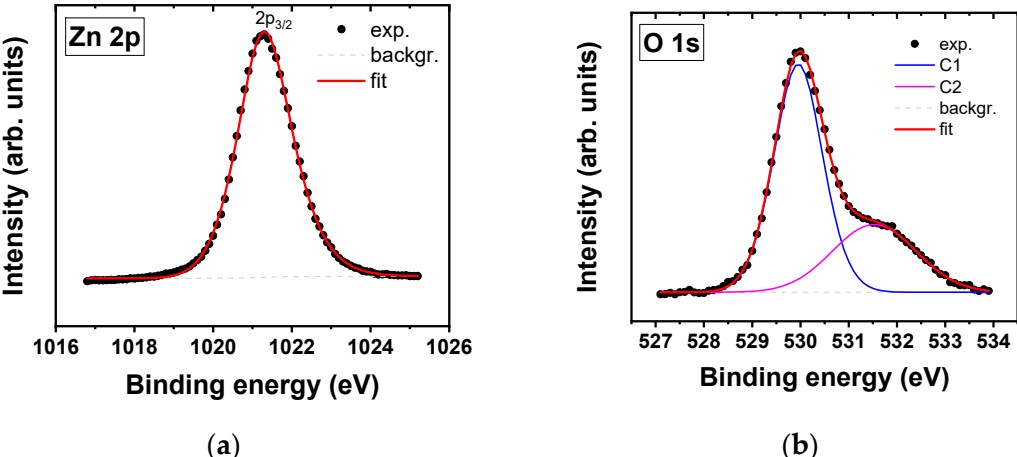

**Figure 3.** (**a**) Zn 2p and (**b**) O1s XPS core lines and respective fits for the undoped ZnO film (NOx).

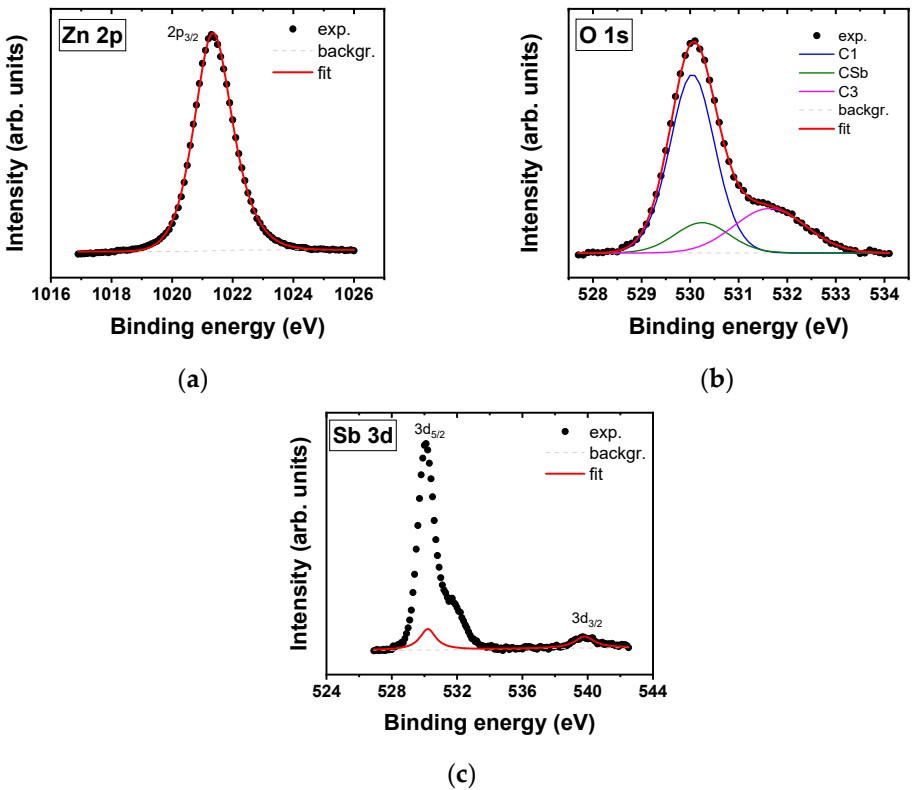

**Figure 4.** (**a**) Zn 2p, (**b**) O1s, and (**c**) Sb 3d XPS core lines and respective fits for the Sb-doped ZnO film (ZnO:Sb21).

**Table 1.** Composition and core line binding energy positions derived from the XPS fits to the ZnO:Sb films data. ΔE is the spin-orbit separation of the Zn 2p and Sb 3d doublets. β is the FWHM of the fitted peaks. A relative area percentage (%) is given for the two O 1s contributions (C1 and C2) with respective FWHM (β1 and β2). CSb in the O1s spectra is attributed to the Sb $3d_{5/2}$ (530.2 eV) doublet peak.

| Sample | Composition (at.%) | | Zn 2p Position /ΔE/ (β) (eV) | O 1s C1 (β1) % CSb (β) % C2 (β2) % (eV) | Sb $3d_{5/2}$ Position /ΔE/ (β) (eV) |
|---|---|---|---|---|---|
| ZnOX | Zn | 49.7 | 1021.3 23.1 (1.7) | 529.9 (1.1) 66% | - |
| | O | 50.3 | | - | |
| | Sb | - | | 531.6 (1.7) 34% | |
| ZnO:Sb21 | Zn | 50.2 | 1021.3 23.1 (1.5) | 530.0 (1.1) 63% | 530.2 9.6 (1.2) |
| | O | 48.8 | | 530.2 (1.2) 13% | |
| | Sb | 1.0 | | 532.0 (1.5) 24% | |

For the XRD analyses, the ZnOX, ZnO:Sb01, ZnO:Sb03, ZnO:Sb07, ZnO:Sb13, and ZnO:Sb21 thin film samples are studied. Figure 5 shows the XRD patterns of these films, for $25° < 2\theta < 65°$, with an incident angle of 1.5° (glancing angle diffraction). The registered diffraction peaks are assigned to a ZnO phase with a hexagonal wurtzite crystal structure, P63mc space group, and the highest intensity peak, ascribed to the (002) atomic planes, which places itself at around 34.5°, corroborated with the information from the crystallographic card 1,011,258 from the ICSD database, served as a study reference. Furthermore, a small peak at ~36° appears for the films with theoretical Sb content (see Figure A4). According to the crystallographic card 1,007,077 from the ICSD database, ascribed to the diamond-like Fd-3m space group, ZnO:Sb presents a peak around 34.6° for the (311) atomic planes and another peak at around 36.1° for the (222) atomic planes. The appearance of a peak at around 36° for the Sb-doped samples, and the increase in its intensity for the films deposited with a greater current applied to the Sb target, corroborates the increasing doping level as ZnO:Sb becomes more present. Added to the doping confirmation, from Figure 5, the highest (002) diffraction peak intensity is associated with the sample deposited with the largest Sb target current density; hence, greater Sb doping improves crystallinity by developing the wurtzite and diamond-like phases. This is also apparent from the SEM cross-section micrographs (see Figure 2), where more highly doped samples present a more regular crystalline column formation and, conversely, undoped ZnO show more three-dimensional crystalline islands.

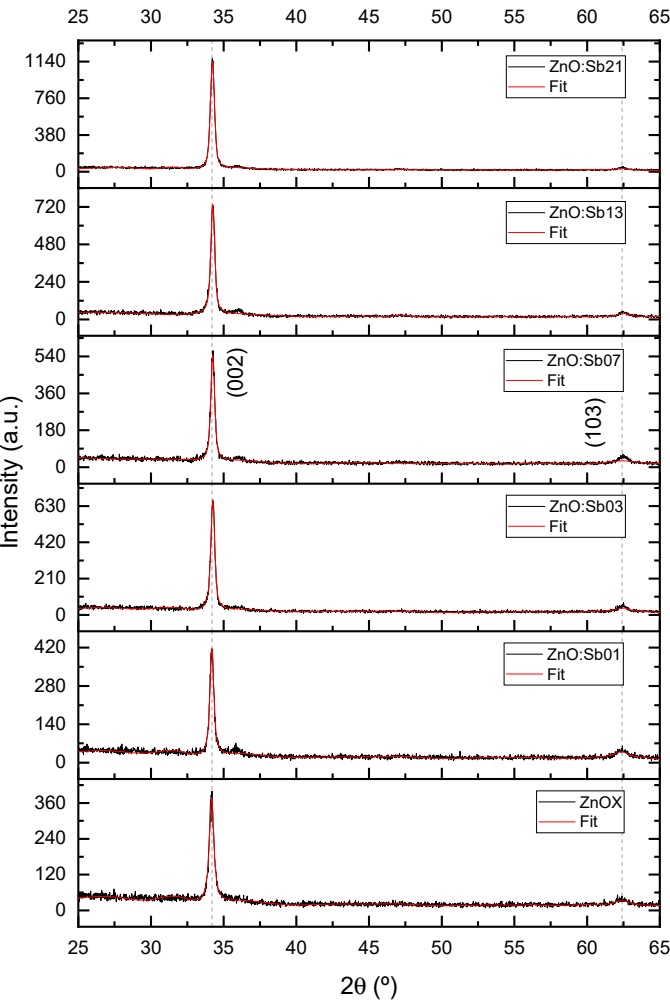

**Figure 5.** XRD patterns with respective line fits for ZnOX, ZnO:Sb01, ZnO:Sb03, ZnO:Sb07, ZnO:Sb13, and ZnO:Sb21 thin films.

The results presented in Table A1 show the lattice parameters (a, c), average crystallite grain size (g) and full width at half maximum (FWHM), that are determined from the XRD analysis and *Fityk* line fits. The lattice parameters are slightly, but not significantly, larger than the reference's parameters of 3.22 Å and 5.20 Å for a and c, respectively, which shows that there is no relevant lattice strain. However, there is a small difference in grain size between the control sample (21 nm) and the doped samples (25 nm), where the grains of the doped samples are slightly larger. This fits well with the previous results, as it also shows the effect and presence of Sb. Furthermore, the uniformity in grain size of all doped samples agrees with the stability of Sb content measured in the XRD analysis.

To more accurately analyze the influence of Sb doping in these properties, the graphs for lattice parameters, grain size, and FWHM, as functions of Sb at.%, are plotted in Figure 6. With these graphs, the data from the samples studied in the EDX analysis is crossed with the data from the XRD analysis, and it is concluded that for the lattice parameters there is no significant change and no obvious correlation that can be stated; however, for the crystallite grain size, there is a tendency for larger Sb content to result in larger crystals.

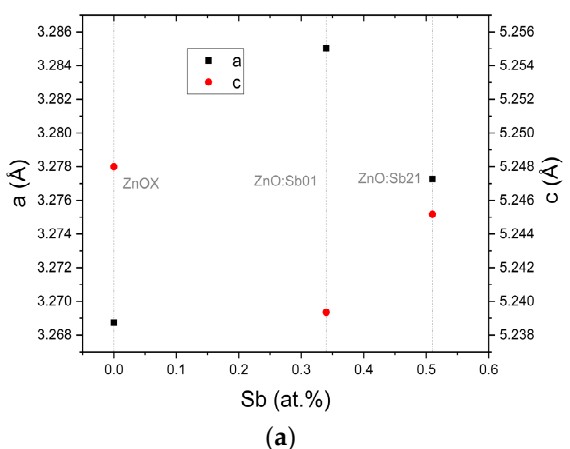
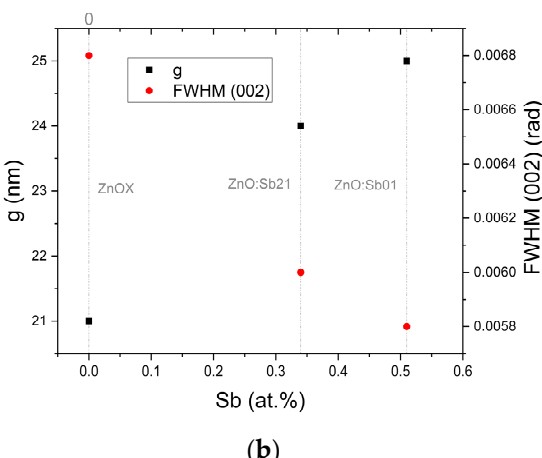

(a)          (b)

**Figure 6.** Plots for (**a**) lattice parameters, and (**b**) crystallite grain size (g) and FWHM of the (002) diffraction peaks as function of the Sb content.

A surface study of the samples is conducted with AFM in a resonance mode of 54 kHz (Figure 7). Table A1 presents the average roughness (Ra) and the quadratic roughness (Rq), obtained with two measurements per sample. The variation of Ra and Rq is plotted in Figure 8 as a function of Sb target current and Sb atomic concentration in the films. Both roughness values are slightly larger for Sb-doped films (9.43 nm and 11.72 nm for Ra and Rq, respectively) when compared with the undoped samples (4.90 nm and 6.35 nm for Ra and Rq, respectively). Its plausible to conclude that Sb increases the roughness of the film, as seen also from the AFM micrographs in Figure 7.

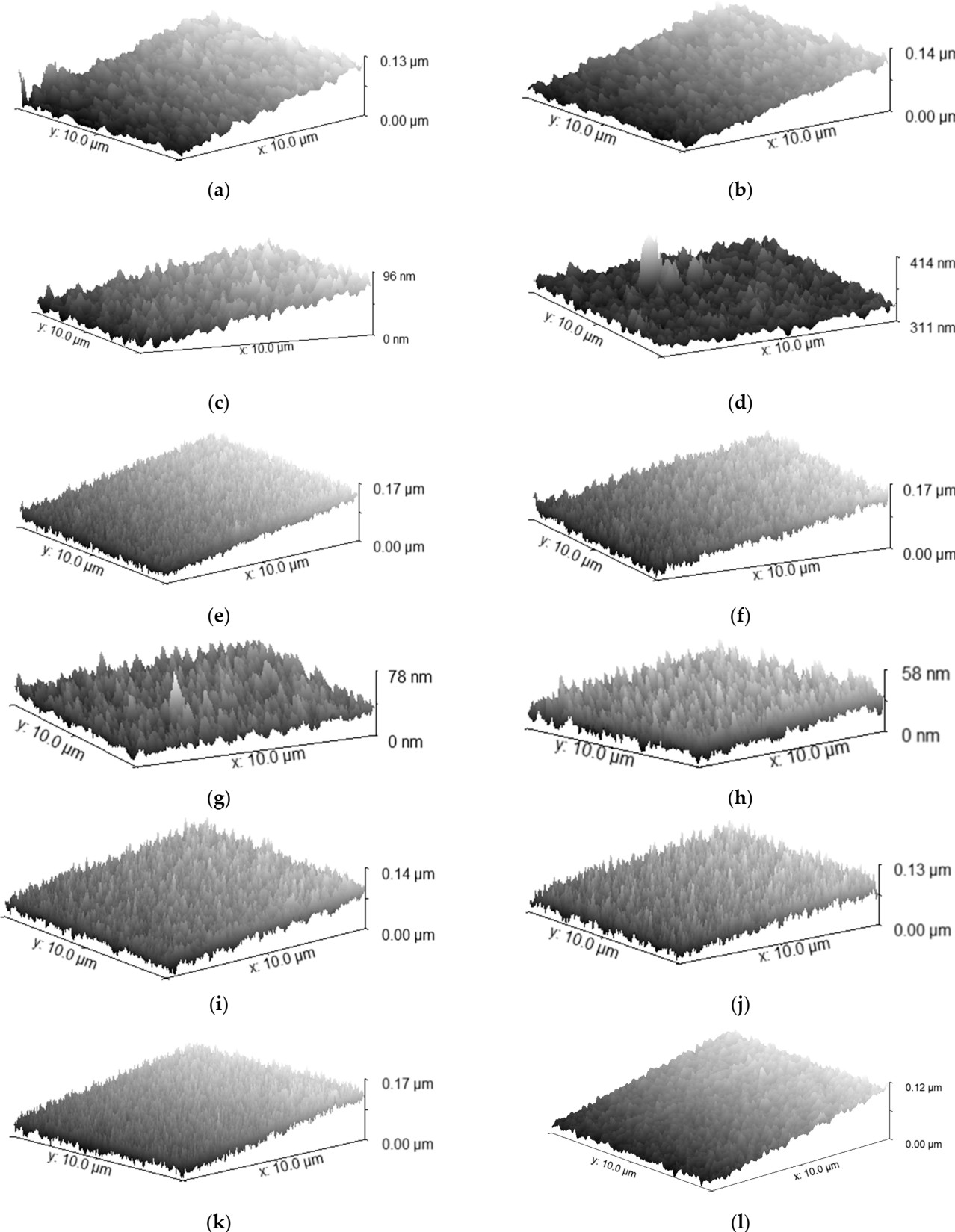

**Figure 7.** AFM micrographs of (**a**,**b**) ZnOX, (**c**,**d**) ZnO:Sb01, (**e**,**f**) ZnO:Sb03, (**g**,**h**) ZnO:Sb07, (**i**,**j**) ZnO:Sb13, and (**k**,**l**) ZnO:Sb21 thin films.

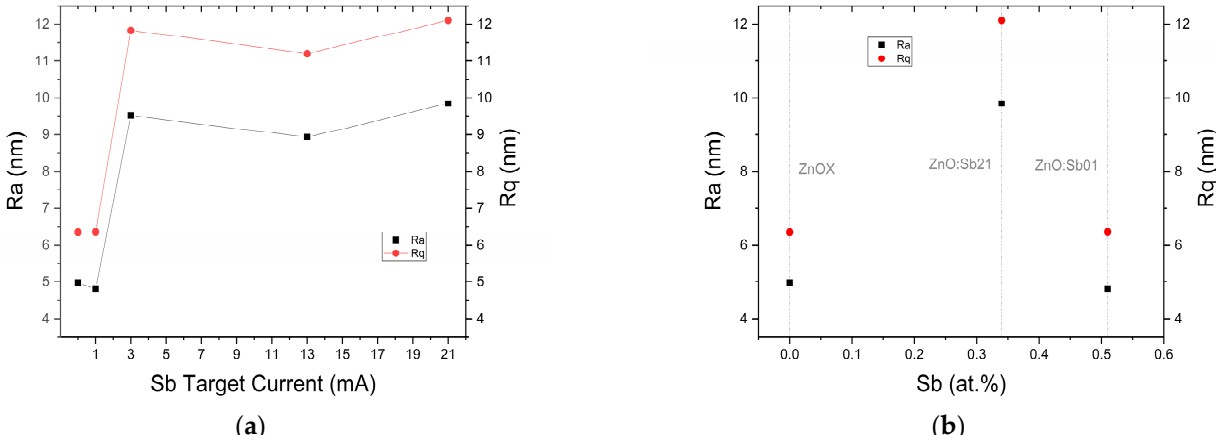

(**a**)
(**b**)

**Figure 8.** Plots for average (Ra) and quadratic roughness (Rq) as functions of (**a**) the target current applied at the Sb target during deposition and (**b**) Sb atomic content.

### 3.3. Grain Boundary Morphology

Figure 9 presents the APT reconstruction of the ZnO:Sb07 thin film, observed from the top (a) and laterally (b,c), relative to the film growth direction, respectively. From this, we observe a clear interface between the Zn (thin film) and Si (substrate) signals and can, therefore, discern heterogeneity in the Zn ions density. Vertical lines can be observed with the Zn-related signals along the film growth, and evidence a grain boundary. This is also observed in a previous study carried out in ZnO:Al and ZnO:Al:Bi samples [34] and results from the columnar structure of the deposited film. Figure 9a displays slices cut to display a 30 nm × 30 nm × 20 nm region in the z-direction, showing a higher Zn content at triple junctions of the grain boundary. The approximate 25 nm grain size agrees with the crystallite size determined from the XRD analysis (Figure 6b).

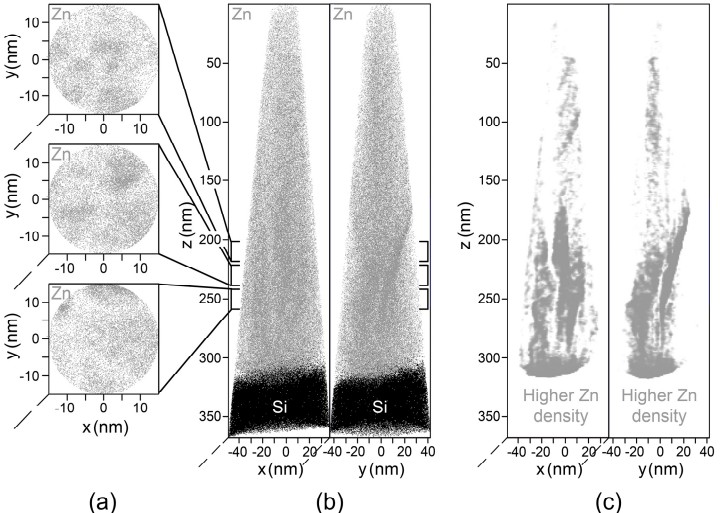

(a)
(b)
(c)

**Figure 9.** Top and side views of a 3D reconstruction obtained by APT of a tip of the ZnO:Sb07 sample. (**a**) Top view sliced section of 30 nm × 30 nm × 20 nm sections; (**b**) side view displaying Zn and Si ions; (**c**) side view of higher density of Zn ions located inside the sample.

Although very little Sb content (~0.01 at.%) is discerned in the reconstruction mass spectrum, a larger Sb inside the sample cannot be excluded (~0.5 at.%). This can be due to the low doping content and difficulty in identifying the Sb-related peaks in the mass spectra, since those can overlap with more significant peaks in the same mass range, such as Zn and Ga.

*3.4. Electrical Properties*

With the Hall measurements, electrical resistivity, carrier concentration, and carrier mobility are determined, and then tabled and plotted in Table A1 and Figure 10. The presence of Sb in ZnO generates a big decrease in the resistivity, which implies that a larger doping level results in a smaller resistivity. The smallest value for resistivity, 0.08 $\Omega \cdot$cm, corresponds to sample ZnO:Sb03, which also shows an increased absolute value of $23.1 \times 10^{18}$ cm$^{-3}$ for carrier concentration and a smaller value for mobility of 1.41 cm$^2 \cdot$V$^{-1} \cdot$s$^{-1}$, properties that are all connected. For carrier mobility, doping causes a slight reduction in its value, while the opposite happens for carrier concentration. More specifically, the carrier mobility of the undoped samples ZnOX and ZnO:Sb00 is 2.57 and 3.46 cm$^2 \cdot$V$^{-1} \cdot$s$^{-1}$, respectively, but the doped samples show smaller values of approximately 1.50 cm$^2 \cdot$V$^{-1} \cdot$s$^{-1}$, with the lowest value of 0.83 cm$^2 \cdot$V$^{-1} \cdot$s$^{-1}$ corresponding to sample ZnO:Sb13. As for carrier concentration, the absolute values of the undoped samples are 4.48 and $2.94 \times 10^{18}$ cm$^{-3}$ for the ZnOX and ZnO:Sb00 samples, respectively, with a significant increase with Sb doping, reaching the largest absolute value of $1.04 \times 10^{20}$ cm$^{-3}$, once again corresponding to the sample ZnO:Sb13. This sample exhibits the normal correlation between these two quantities, as the increase in carrier concentration is associated with a decrease in mobility. Furthermore, this sample also shows a reduced electrical resistivity of 0.18 $\Omega \cdot$cm; although this does not represent the smallest value, it is more than likely linked to these other electrical properties. Finally, the negative nature of the carrier concentration shows that the films have *n*-type conductivity, preserving this inherent characteristic of the ZnO.

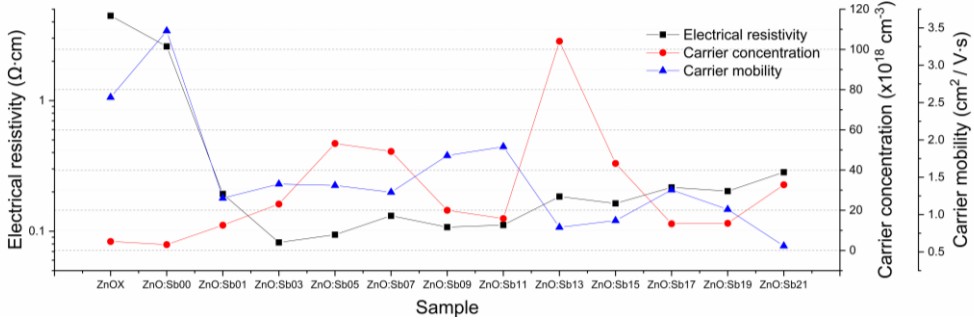

**Figure 10.** Electrical resistivity and carrier concentration and mobility at RT as functions of the sample.

*3.5. Thermoelectricity Analysis*

As expected, the control sample and the ZnO:Sb00 sample have significantly small absolute values of the Seebeck coefficient, at 2.8 and 6.5 µV/K, respectively. However, this property quickly rises for the other films, with small exceptions. In most cases it reaches an absolute value close to 100 µV/K, with the greatest value of 100.4 µV/K corresponding to the ZnO:Sb13. As is observable in Figure 11, the Seebeck coefficient stabilizes for larger currents. Furthermore, its negative nature corroborates with what was previously concluded, that the films display *n*-type conductivity. When associated with electrical conductivity, the films showcase interesting values for PF, with the greatest being 1.1 mW$\cdot$m$^{-1} \cdot$K$^{-2}$ at 300 K, which is much larger than all the values found in the literature for ZnO:Sb films. Previous work on ZnO:Sb films deposited using the same deposition system but not in a confocal geometry, with Sb pellets on the ZnO target, resulted in a much lower PF of 0.2 µW$\cdot$m$^{-1} \cdot$K$^{-2}$ at 300 K [35]. In this previous work, the reproducibility of the films is more difficult due to accelerated erosion of the Sb pellets during sputtering, in contrast with the confocal geometry of this present work, where the current density applied to the Sb target is adequately controlled and independent of the ZnO target power. As expected, the largest PF value corresponds to the sample with the greatest absolute Seebeck coefficient (ZnO:Sb13). However, the latter sample also shows the largest absolute value for carrier concentration ($1.04 \times 10^{20}$ cm$^{-3}$); this and other beneficial electrical properties,

such as resistivity, also influenced the PF value of the sample, making it the most successful film fabricated and the best example of the positive effect of Sb doping in terms of thermoelectric performance.

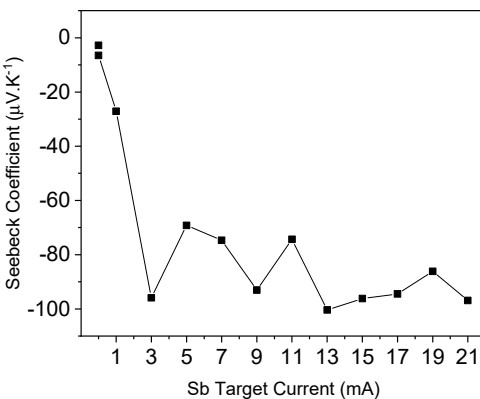

**Figure 11.** Seebeck coefficient at RT as a function of Sb target current.

## 4. Conclusions

The goals of this work are to perform a complete evaluation of the effects of Sb doping in optically transparent ZnO thin films, and to ascertain the relevance of this material for thermoelectric applications, more specifically TCO technology. Several samples were fabricated with variating deposition parameters, namely the induced current in the doping target of Sb, to study the effect of this dopant on the film properties.

The produced ZnO:Sb thin films range in thickness between 600 and 800 nm and share an energy band gap of 3.25 eV, without relevant variations, and high average transmittance, above 80%. Before the depositions, the Sb content was expected to increase alongside the induced current in the Sb target, but this is not verified. Instead, a stagnant doping value (~0.5 at.% of Sb) is obtained, probably resulting from an unsurpassable threshold for Sb doping in the ZnO cell upon co-sputtering in a confocal geometry; however, further studies will be required to confirm this. Nevertheless, the presence of Sb is verified, although in small quantities, and its effect on the films' properties is prevalent. The doped thin films have significantly smaller electrical resistivity values, with the smallest being 0.08 $\Omega$·cm, and larger carrier concentrations, with the largest absolute value of $1.0 \times 10^{20}$ cm$^{-3}$, which allows to conclude that the produced films are significantly enhanced in electrical conductivity due to the presence of Sb. Additionally, since the Sb content is very low, not surpassing 0.51 at.% in the EDX analysis and only showing 1.0 at.% (at the surface) from the XPS analysis, this still has a relatively high effect on the conductive properties of the samples. It is reasonable to assume that, if it is possible to increase Sb doping, then the electrical conductivity of the film will increase. Furthermore, the films achieve an excellent thermoelectric performance for an Sb target current of 13 mA (current density of 0.17 mA/cm$^2$), by attaining the largest absolute Seebeck coefficient of 100 $\mu$V/K, and a respective PF with an impressive value of 1.1 mW·m$^{-1}$·K$^{-2}$ at 300 K. Once again, these values are obtained with a low Sb content (~0.5 at.%). Thus, this doping process has the potential for better results. Two of the three conditions for producing a good thermoelectric material are fulfilled, namely enhancing electrical conductivity and improving the Seebeck coefficient. The missing condition is the thermal conductivity enhancement, which requires additional experiments, currently being designed. Moreover, the optical transparency remains with the doping; hence, it is proven that Sb doping of ZnO is a good and promising solution for TCO-based applications. From atom probe tomography experiments, a more significant Zn content is registered on triple junctions at the grain boundary.

Finally, considering all the properties previously mentioned, several ZnO:Sb films with good characteristics are developed, and the results testify the potential of this material

in practical applications as optically transparent thermoelectric thin films. From these appealing results, it would be possible to implement these ZnO:Sb films in electronic devices and, perhaps one day, to use this material in display screens and windows of skyscrapers for thermal heat harvesting, positively contributing to solving the emerging global energy problem.

**Author Contributions:** Conceptualization, C.J.T.; methodology, C.J.T.; software, C.J.T., J.M.R. and H.F.F.; validation, C.J.T. and T.B.; investigation, C.J.T., J.M.R. and H.F.F.; resources, C.J.T. and T.B.; data curation, J.M.R. and H.F.F.; writing—original draft preparation, H.F.F.; writing—review and editing, C.J.T. and J.M.R.; supervision, C.J.T.; project administration, C.J.T. and T.B.; funding acquisition, C.J.T. and T.B. All authors have read and agreed to the published version of the manuscript.

**Funding:** This research is funded by FCT/PIDDAC through the Strategic Funds project reference UIDB/04650/2020-2023. Joana M. Ribeiro is grateful to the Fundação para a Ciência e Tecnologia (FCT, Portugal) for the Ph.D grant SFRH/BD/147221/2019. This work (proposal ID 2021-025-030112) was carried out with the support of the Karlsruhe Nano Micro Facility (KNMFi, www.knmf.kit.edu, accessed on 8 March 2023), a Helmholtz Research Infrastructure at Karlsruhe Institute of Technology (KIT, www.kit.edu, accessed on 8 March 2023).

**Institutional Review Board Statement:** Not applicable.

**Informed Consent Statement:** Not applicable.

**Data Availability Statement:** Data can be shared upon formal request.

**Conflicts of Interest:** The authors declare no conflict of interest.

## Appendix A

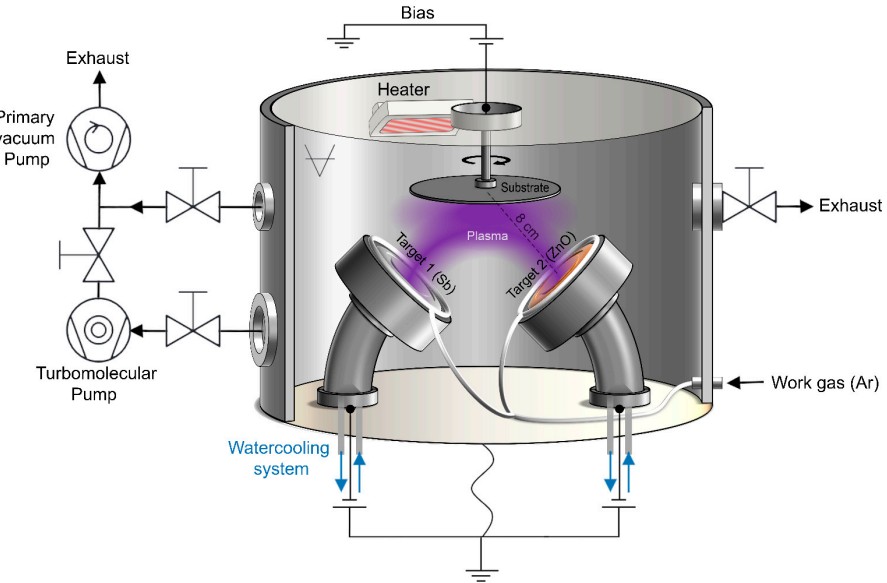

**Figure A1.** Schematic of the sputtering deposition chamber.

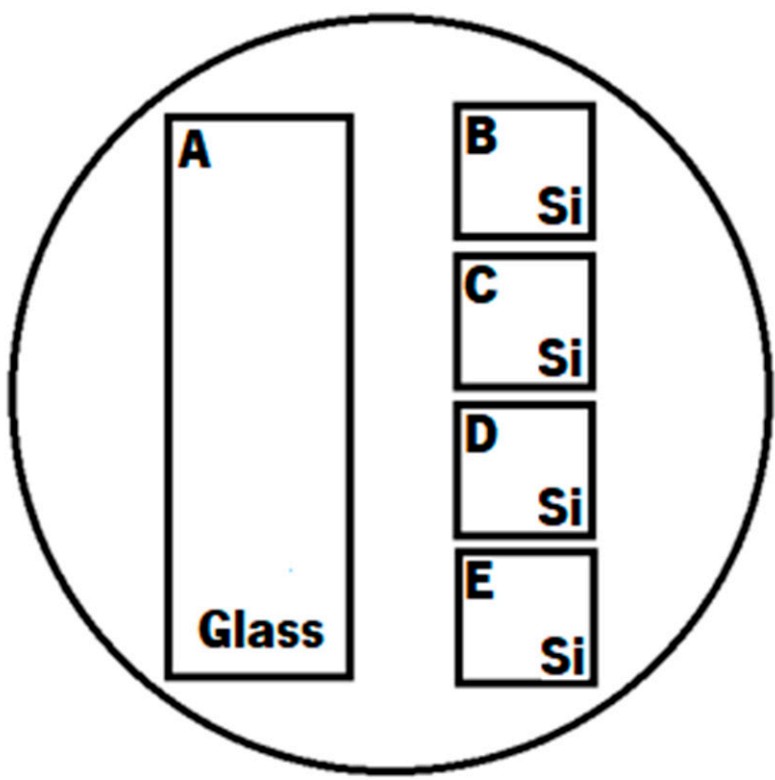

**Figure A2.** Layout of the substrate holder.

**Appendix B**

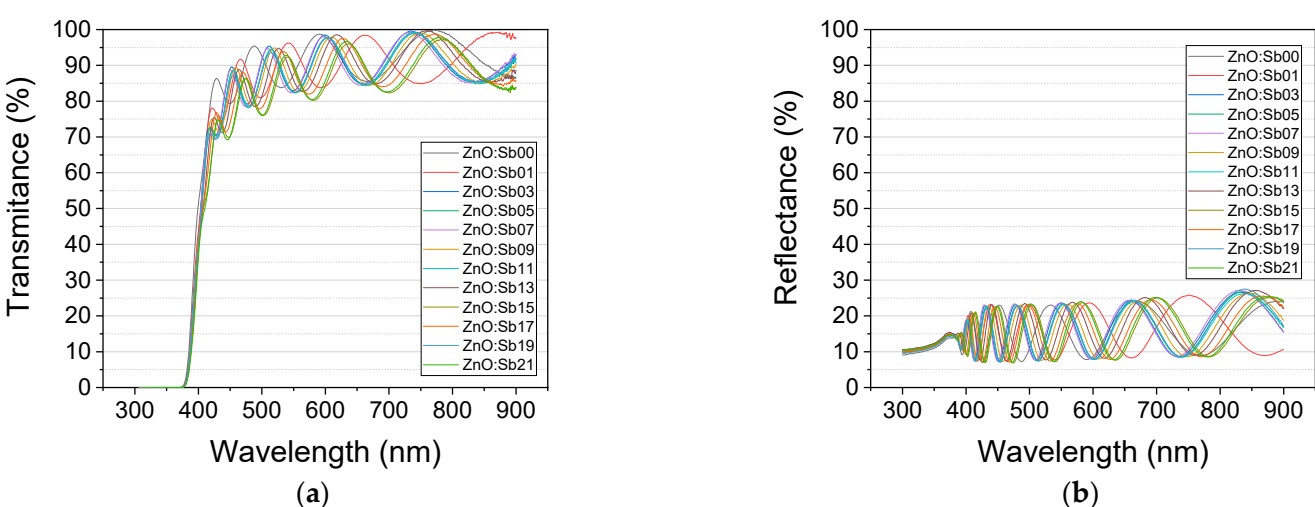

(**a**)

(**b**)

**Figure A3.** Optical (**a**) transmittance and (**b**) reflectance of each sample.

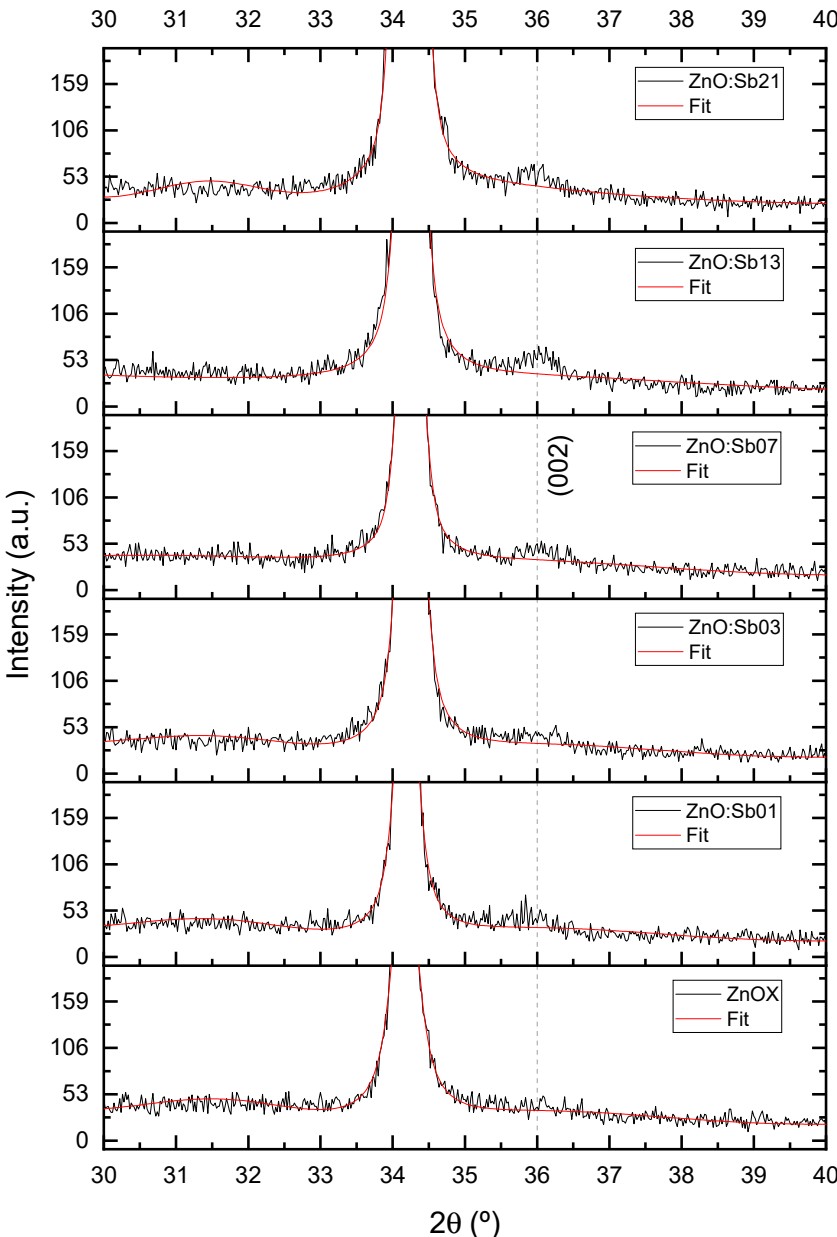

**Figure A4.** Zoomed in XRD patterns and respective line fits for ZnOX, ZnO:Sb01, ZnO:Sb03, ZnO:Sb07, ZnO:Sb13, and ZnO:Sb21 films to highlight the small peak at around 36° that is ascribed to the Sb (002) atomic planes, which is not easily discerned in the full XRD patterns of Figure 5.

## Appendix C

**Table A1.** Samples and respective important properties, absolute Seebeck coefficient |S|, electrical resistivity ($\rho$), carrier concentration ($n$), carrier mobility ($\mu$), lattice parameters (a, c), grain size (g) mean, and quadratic roughness (Ra, Rq). Thickness calculated and measured by SEM analysis (t(SEM)), Sb content measured by EDX analysis (Sb(EDX)), average transmittance and reflectance ($\overline{\mathrm{T}}, \overline{\mathrm{R}}$), and energy band gap (Eg).

| Sample | \|S\| ($\mu$V/K) | PF (mW·m$^{-1}$·K$^{-2}$) | $\rho$ ($\Omega$·cm) | $n$ ($\times 10^{18}$cm$^{-3}$) | $\mu$ (cm$^2$/V·s) | a (Å) | c (Å) | g (nm) | Ra (nm) | Rq (nm) | t(SEM) (nm) | Sb(EDX) (at.%) | $\overline{\mathrm{T}}$ * (%) | $\overline{\mathrm{R}}$ * (%) | Eg (eV) |
|---|---|---|---|---|---|---|---|---|---|---|---|---|---|---|---|
| ZnOX | 2.8 | $0.2 \times 10^{-4}$ | 4.46 | −4.5 | 2.6 | 3.27 | 5.25 | 21 | 4.97 | 6.35 | 578.47 | - | 87.41 | 16.74 | 3.26 |
| Zno:Sb00 | 6.5 | $0.2 \times 10^{-3}$ | 2.60 | −2.9 | 3.46 | - | - | - | - | - | 627.03 | - | 87.50 | 16.54 | 3.26 |
| ZnoZnO:Sb01 | 27.1 | $0.4 \times 10^{-2}$ | 0.19 | −12.6 | 1.22 | 3.28 | 5.25 | 25 | 4.82 | 6.35 | 702.30 | 0.51 | 87.01 | 15.21 | 3.25 |
| ZnO:Sb03 | 95.9 | 1.1 | 0.08 | −23.1 | 1.4 | 3.28 | 5.24 | 25 | 9.52 | 11.84 | 719.01 | - | 85.83 | 16.54 | 3.25 |
| ZnO:Sb05 | 69.2 | 0.5 | 0.09 | −53.3 | 1.4 | - | - | - | - | - | 740.74 | - | 85.00 | 16.45 | 3.25 |
| ZnO:Sb07 | 74.7 | 0.4 | 0.13 | −49.3 | 1.3 | 3.29 | 5.24 | 26 | 5.23 | 6.59 | 757.60 | 0.50 | 85.35 | 16.69 | 3.24 |
| ZnO:Sb09 | 93.0 | 0.8 | 0.11 | −20.0 | 1.8 | - | - | - | - | - | 771.38 | - | 85.03 | 16.42 | 3.26 |
| ZnO:Sb11 | 74.4 | 0.5 | 0.11 | −15.9 | 1.9 | - | - | - | - | - | 778.07 | 0.12 | 84.83 | 16.33 | 3.25 |
| ZnO:Sb13 | 100.4 | 0.5 | 0.18 | −104.1 | 0.8 | 3.27 | 5.24 | 25 | 8.93 | 11.21 | 782.57 | - | 84.59 | 16.69 | 3.25 |
| ZnO:Sb15 | 96.2 | 0.6 | 0.16 | −43.3 | 0.9 | - | - | - | - | - | 785.20 | - | 81.80 | 15.65 | 3.24 |
| ZnO:Sb17 | 94.5 | 0.4 | 0.22 | −13.3 | 1.3 | - | - | - | - | - | 787.07 | - | 83.69 | 15.99 | 3.25 |
| ZnO:Sb19 | 86.1 | 0.4 | 0.20 | −13.5 | 1.1 | - | - | - | - | - | 788.52 | - | 84.66 | 16.39 | 3.25 |
| ZnO:Sb21 | 96.9 | 0.3 | 0.28 | −32.7 | 0.6 | 3.29 | 5.24 | 24 | 9.84 | 12.10 | 789.70 | 0.34 | 81.91 | 15.93 | 3.24 |

* Average optical transmittance and reflectance calculated between 400 and 700 nm.

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
