# Peer review of "Thermoelectric and Structural Properties of Transparent Sb-Doped ZnO Thin Films Sputtered in a Confocal Geometry"

_coatings, doi:10.3390/coatings13040735_

Round 1
Reviewer 1 Report
Comments and Suggestions for Authors
In this work entitled “Thermoelectric properties of transparent Sb-doped ZnO thin films” by H.F. Faria et al, the study is based on the optical, morphological and thermoelectrical properties of co-sputtered ZnO:Sb thin films in confocal geometry under varied induced current on Sb target. Optical, morphological, topographical and compositional properties of a 30-minute ZnO:Sb film deposited at 0 to 21 mA.cm2 Sb target current and 10 mA.cm2 ZnO target current. Optical analysis confirmed the presence of Sb had almost no effect on the transparency of ZnO films. The results of SEM images and EDX on five distinct samples indicated that all samples contained around 0.5 at% Sb and that a higher current density in the Sb target resulted in films with a higher thickness, ranging from 578 nm to 790 nm. For the XRD analyses, ICSD card no: 1011258 for hexagonal ZnO crystals is derived. XPS analysis are performed on ZnO baseline sample as well as ZnO:Sb21. AFM profiles demonstrated an increase in Ra with Sb target current and APT showed very little Sb content (~0.01 at.%) discerned in the reconstruction mass spectrum. Hall measurement also is conducted for all the samples and showed smaller values of approximately 1.50 cm2∙V-1∙s-1, with the lowest value of 0.83 cm2∙V-1∙s-1 corresponding to sample ZnO:Sb13. Ultimately, thermoelectrical test conducted and Sb dopped films demonstrated remarkable thermoelectric performance, with the ZnO:Sb13 film achieving the greatest absolute Seebeck coefficient of 100 V/K and a PF of 1.1 mWm-1K-2, concluding that potential of Sb doping. Overall, ZnO based thin films have seen a spike in application for their inherent thermoelectrical potentials. Yet, this study lacked highlighting the significant novelty it offers compared to earlier reports, which resulted in a general conclusion that was insufficiently credible as new research.
Therefore, I would recommend publication pending major revisions:
1. The title is too general, you may consider “Thermoelectric characteristics of sputtered AZO thin films with varying current density” or a more relevant title.
2. In the Abstract, the applied current variation range should be stated.
3. In the introduction, please limit the renewable energy fundamentals (line 23 to 40) to two or maximum three sentences.
3.1. Add references for Seebeck definition (line 47).
3.2. Add reference for all the presented equations in this manuscript.
4. In the introduction, line 57 to 72 have minor relevance to the presented work, you may consider adding literature on various ZnO doping methods by different elements (Al, Ga, Bi,..etc.) that showed a potential improvement or address the previously reported thermoelectric properties of ZnO.
5. Lines 74 to 90 are on the prospects of TCOs; please summarize it and instead elaborate on the recent issues in ZnO thermoelectrical enhancements and the gaps that the current work aims to address.
6. Line 104 to 119 can be omitted/ Since p-type ZnO is not the focus of this work.
7. In the introduction, if the major improvement in this study is the confocal sputtering system, please explain the substantial limitations in previously used deposition techniques that made confocal sputtering a new alternative.
8. Section 3.1, line 295, “further increase in the Sb target current density in confocal geometry does not overcome this threshold.” Please provide a reference to support this remark. If any other studies using confocal configurations encountered growth/doping limitations.
9. Section 3.2, line 334, Is there any study that reported the same ICSD card (1011258) while doping Sb? please include supporting references.
10. Section 3.2, line 338 to 341, Is there a literature that confirms introducing impurity to the crystal structure enhances the host's crystallinity? please elaborate and include supporting references.
11. Section 3.2, line 344, use a more practical phrase to explain what “better cross-section crystallinity” entails.
12. Section 3.2, consider presenting Figure 8 before the summarized graphs in Figure 7. (Easier to follow)
13. Section 3.5, please add earlier Sb doping methods as well as Seebeck findings from other ZnO doped research and compare the results for a more coherent conclusion.
14. In conclusion, include the optimal current in this study.
Thanks.

Author Response
The authors are grateful for the reviewer's comments.

Reviewer 2 Report
This manuscript presents a set of experiments synthesizing and characterizing Sb-doped ZnO thin films deposited on a Si substrate. The authors have varied the Sb concentration in their samples while keeping all other synthesis parameters constant, isolating the effect of Sb dopants on the optical, transport, and thermoelectric properties of the film. Oxide thermoelectric materials have great potential as they can be easily synthesized at scale, so this line of inquiry deserves further investigation. As a result, I can recommend this manuscript for publication after the authors fully elaborate on the following points required for a better understanding of the work:
1. It is still not clear what oxidation state Sb dopants adopt. Sb can be either 3+ or 5+. Answering this question may also address how the incorporation of Sb flips the ZnO films from Zn deficient to O deficient.
2. The exact measurement temperatures should be reflected in the captions of Figures 10 and 11.
3. Sb dopants, in particular, have been predicted to be efficient in controlling the thermoelectric properties of oxides. Can authors draw parallels? Assadi et al. Magnetic, electrochemical and thermoelectric properties of P2-Nax(Co7/8Sb1/8)O2, Chemical Physics Letters, 687 (2017) 233-237; https://doi.org/10.1016/j.cplett.2017.09.026
4. A comprehensive review outlining the importance of thermoelectric oxides deserves mention in the introduction. Most of the introduction, in its current form, is dedicated to the fundamentals of the thermoelectric theory and the doping characteristics of ZnO. It would be beneficial to the reader if the introduction could specifically address the suitability of oxide thermoelectric materials over other common materials classes, such as non-toxicity, facile synthesizability, and being composed of earth-abundant materials: Yin et al. Recent advances in oxide thermoelectric materials and modules, Vacuum, 146, (2017) 356-374; https://doi.org/10.1016/j.vacuum.2017.04.015
Author Response

(The authors gave the same response as above.)

Reviewer 3 Report
Referee report on “Thermoelectric properties of transparent Sb-doped ZnO thin films”
This is a rather interesting and good paper that certainly can probably be recommended for publication, but clarifying and detailing some parts of the text.
1. Imtroduction. 2nd paragraph. This is an introduction of the mamuscript and the absence of the supporting references is not good for acquainting readers with the essence of the problem under consideration.
2. Line 79. “The most utilized TCO is tin-doped indium oxide (ITO) [21]” Note that [21] reference is of 2005 and does not reflect what has been done in the last 18 years. For recent results, please see recent MDPI paper:
Almaev, A.V.; et al . ITO Thin Films for Low-Resistance Gas Sensors. Materials 2023, 16, 342. https://doi.org/10.3390/ma16010342
3. Speaking of Sb doping, in what valence state does the impurity enter, and understanding that its valency differs from that of Zn2+, is a vacancy required for charge compensation?
4. Line 256. Must be ZnO:SbXX.
5. Line 283-285. Does it mean that such small impurity concentrations change the width of the band gap Eg or does the impurity simply give its additional level at the edge of the zone? See about this problem, recent commentary paper of Editors of “Optical Materials”: Brik, M. G., et al (2022). A few common misconceptions in the interpretation of experimental spectroscopic data. Optical Materials, 127, 112276.
https://doi.org/10.1016/j.optmat.2022.112276
6. Fig. 2 The legend in the figures is not distinguishable and needs to be improved.
7. Fig. 6. The statement about oxygen vacancies would require additional measurements, such as photoluminescence, where all types of vacancies can be clearly distinguished.
See, for example: Uklein, A. V., et al (2018). Nonlinear optical response of bulk ZnO crystals with different content of intrinsic defects. Optical Materials, 84, 738-747.
8. Was the stability/aging of the synthesized samples investigated/verified?
9. In the conclusions, it is necessary to clearly formulate what new data about the studied materials were obtained/suggested in this work?
In general, the manuscript is interesting and can be recommended for publication after constructive reflection on the above comments.
Author Response

(The authors gave the same response as above.)

Round 2
Reviewer 1 Report
I recommend publication in present form.
Thanks
Author Response
The authors are very grateful for the review.
Reviewer 2 Report
In light of revisions, this manuscript can be accepted.
Author Response

(The authors gave the same response as above.)

Reviewer 3 Report
The following questions were not answered or answered uncorrectly.
1. “ Speaking of Sb doping, in what valence state does the impurity enter, and understanding that its valency differs from that of Zn2+, is a vacancy required for charge compensation? “ To continue,
I must to note that each Sb3+ ion requires charge compensation as a cation vacancy.
This is very classical case, when charge compensation is needed. For example in the case of KCl, two cation vacancies are required: Aguado, A. (2002). Lattice distortions induced by As3+, Sb3+, and Bi3+ substitutional impurities in KCl: An embedded cluster study. The Journal of Physical Chemistry B, 106(28), 6991-6996.
Note, that Sb3+ ions produce their own absorption band, and there are a lot of papers about this properties. See, just one example:
Tsuboi, T., Ahmet, P., & Kang, J. G. (1992). Optical absorption bands due to the s2 to sp transition in KCl: Sb3+ crystals. Journal of Physics: Condensed Matter, 4(2), 531.
Note that in all cases when the absorption bands of Sb3+ were established, Eg does not change.
This is why, the authors answered the following question incorrectly, too, namely:
2. Line 283-285. Does it mean that such small impurity concentrations change the width of the band gap Eg or does the impurity simply give its additional level at the edge of the zone? See about this problem, recent commentary paper of Editors of “Optical Materials”: Brik, M. G., et al (2022). A few common misconceptions in the interpretation of experimental spectroscopic data. Optical Materials, 127, 112276. https://doi.org/10.1016/j.optmat.2022.112276
More note: Author have mentioned in the paper (Ribeiro et al all, Journal of Alloys and Compounds. Volume 838, 15 October 2020, 155561). Fig. 12. in that paper gives more questions than answers. But what is clear, EXAFS will not help to find proper value of Eg. Additional note, that the mention of EXAFS here is not only inappropriate, but also not smart. EXAFS people (A. Kuzmin) do know this fact very well. For example, the same KCl mentioned above, pure, or with a vacancy concentration of 1019 cm-3, will be the same for EXAFS. EXHAFS, as a structural method, does not see such concentrations.
Author Response
Again, we thank the reviewer for his comments.
The authors would like to point out that perhaps we were misunderstood. As stated in the previous reply, combined EXAFS/XANES experiments are planned, dependent on project approval for beam time in the synchrotron, in order to study the local order and the Sb valence state. The authors did not imply that EXAFS would be used to determine the band-gap. We apologize for this misunderstanding. But these experiments will also be useful for determining the valence state and corroborating with the present XPS findings reported in the manuscript.
We agree with the reviewer that it is important to determine the band gap and we have determined it by using the optical data, being the variation within the films is very small, as stated in the manuscript. Further experiments are planned to use ultraviolet photoelectron spectroscopy (UPS). Although UPS will not determine the band gap directly, possibly the Fermi level for undoped and doped films, which will shine some light on what the reviewer mentioned regarding the displacement of the conduction and valence bands due to impurity doping.
Regarding the Sb3+ optical absorption bands, in particular, those arising from charge compensation, the authors performed experiments at RT and were unable to see differences between spectra of undoped and Sb-doped ZnO.
Round 3
Reviewer 3 Report
Dear Authors,
1. In your last reply, you stated: “Regarding the Sb3+ optical absorption bands, in particular, those arising from charge compensation, the authors performed experiments at RT and were unable to see differences between spectra of undoped and Sb-doped ZnO “
However, in you manuscript, lines 257-258, we can find: “The effect of the Sb-doping was reflected in this analysis as a decrease from 3.37 eV of the undoped ZnO to 3.25 eV for Sb-doped ZnO was registered”
Please reply, where are you telling the truth and where are you misleading?
2. Concerning sentence: “The effect of the Sb-doping was reflected in this analysis as a decrease from 3.37 eV of the undoped ZnO to 3.25 eV for Sb-doped ZnO was registered”
This is understandable behavior, because doping with Sb3+ requires cation vacancies as a charge compensator. These cationic vacancies create additional absorption at the absorption edge, which leads to a seemingly visible change in the absorption edge. This situation has been explained in detail in a dedicated article, recently written by the Editors of “Optical Materials” (Elsevier) journal: Brik, M. G., Srivastava, A. M. (2022). A few common misconceptions in the interpretation of experimental spectroscopic data. Optical Materials, 127, 112276.
3. Why are the bands associated with Sb3+ not visible? To answer, it is important to remember, that Eg of Sb2O3 is 4.3- 4.4 eV, which indicates that the absorption bands of Sb3+ are greater than Eg in ZnO. ( Zhang, Jinzhong, et al. "Thermally robust optical properties of the wafer-scale α-Sb2O3 films." Applied Surface Science 612 (2023): 155793 )
https://doi.org/10.1016/j.apsusc.2022.155793
In the conclusion, this manuscript can be considered for publication after constructive reflection on the above comments.
Author Response
Dear Authors,
- In your last reply, you stated: “Regarding the Sb3+ optical absorption bands, in particular, those arising from charge compensation, the authors performed experiments at RT and were unable to see differences between spectra of undoped and Sb-doped ZnO “
However, in you manuscript, lines 257-258, we can find: “The effect of the Sb-doping was reflected in this analysis as a decrease from 3.37 eV of the undoped ZnO to 3.25 eV for Sb-doped ZnO was registered”
Please reply, where are you telling the truth and where are you misleading?
Reply: The authors were referring to hypothetical absorption bands characteristic to the Sb3+ ions, in the UVC region; of course changes were observed in the absorption edge, as mentioned in the band gap discussion in the manuscript. Since our films are deposited on glass, this substrate absorbs most of the UVB and UVC; hence, it is not possible to survey this part of the absorption spectra. For example, in the case that the reviewer mentioned of KCl: Sb3+ (Tsuboi, 1992), the materials are in the form crystals, and hence the absorption can be readily monitored using an UV-Vis spectrophotometer. In order to circumvent this problem, the films would have to be deposited on a substrate that could be dissolved, for example NaCl in water, although NaCl has a strong absorption band around 200 nm, and the monocrystal epitaxy of the substrate may induce a different growth of ZnO:Sb when compared to deposition on glass. The alternative is to perform an in-depth spectroscopic ellipsometry study, for example in the vein of what was performed by Zhang (2022) on α-Sb2O3 films, that the reviewer reported. At the moment, the authors do not have access to this type of equipment. A collaboration would have to be procured and inevitably this takes time, study the literature, prepare the films, choose the optimal model, do the experiments, and analyse the data. The authors agree that this needs to be done, as the UPS and EXAFS experiments mentioned in the last review round, and possibly PL also to study the origin of oxygen vacancies created for charge compensation. We hope to perform this study and publish in due time in a new publication.
The corresponding author would like to ask the reviewer to see, for example, chronologically the evolution of the studies that this author performed for other metal-oxide systems with thermoelectric properties, such as ZnO:Ga,Al,Bi and TiO2:Nb, where besides electrical and thermoelectric studies, an in-depth study on the composition and atomic environment was performed, for example. The authors do not slice the data for different publications. All papers are published once new interesting data is compiled to continue the study of a chosen material. This is also the case for the current ZnO:Sb films, which is not certainly closed, and more studies are in progress, and naturally take time. For this manuscript the relevance goes to the thermoelectric properties analysis.
- Concerning sentence: “The effect of the Sb-doping was reflected in this analysis as a decrease from 3.37 eV of the undoped ZnO to 3.25 eV for Sb-doped ZnO was registered”
This is understandable behavior, because doping with Sb3+ requires cation vacancies as a charge compensator. These cationic vacancies create additional absorption at the absorption edge, which leads to a seemingly visible change in the absorption edge. This situation has been explained in detail in a dedicated article, recently written by the Editors of “Optical Materials” (Elsevier) journal: Brik, M. G., Srivastava, A. M. (2022). A few common misconceptions in the interpretation of experimental spectroscopic data. Optical Materials, 127, 112276.
Reply: The authors strongly agree, as stated in the previous point.
- Why are the bands associated with Sb3+not visible? To answer, it is important to remember, that Eg of Sb2O3 is 4.3- 4.4 eV, which indicates that the absorption bands of Sb3+ are greater than Eg in ZnO. ( Zhang, Jinzhong, et al. "Thermally robust optical properties of the wafer-scale α-Sb2O3 films." Applied Surface Science 612 (2023): 155793 )
https://doi.org/10.1016/j.apsusc.2022.155793
Reply: The authors are aware of this and have thoroughly explained the inherent problem in point 1.
In the conclusion, this manuscript can be considered for publication after constructive reflection on the above comments.
Reply: The authors are committed to perform a more in-depth study on the optical absorption properties of these films, their atomic environment and band structure. But, as the Reviewer certainly knows, this takes time. Based on this explanation, we kindly request the Reviewer to accept the manuscript as it is knowing that the authors are committed to pursue these studies that the reviewre suggested and the authors are aware of their importance.
